# Burden, risk factors, and forecasts of gout in BRICS countries, 1990–2021: Insights from the Global Burden of Disease Study 2021

**Qizhou Mo**[1☯], **Shufeng Luo**[2☯], **Fengyi Wang**[3☯], **Haiqi Liang**[1], **Jiayin Yu**[1], **Naikai Liao**[1], **Min Qin**[4*], **Jiwen Cheng**[1*]

1 Department of Urology, The First Affiliated Hospital of Guangxi Medical University, Nanning, China, 2 Department of Urology, Red Cross Hospital of Yulin City, Yulin, China, 3 Graduate School of Guangxi Medical University, Nanning, China, 4 Human Sperm Bank, The First Affiliated Hospital of Guangxi Medical University, Nanning, China

☯ These authors contributed equally to this work
* chengjiwen@stu.gxmu.edu.cn; 63086204@qq.com

## Abstract

### Objectives

To evaluate the temporal trends and projected burden of gout in BRICS (Brazil, Russia, India, China, and South Africa) countries from 1990 to 2021, based on Global Burden of Disease 2021 data, and to explore the contributions of key risk factors.

### Methods

Age-standardized prevalence rates (ASPR), years lived with disability rates (ASYR), and case counts were extracted from the GBD Results Tool. Temporal trends were assessed using estimated annual percentage change (EAPC). Joinpoint regression evaluated time-varying changes in ASYR attributable to high BMI (Body Mass Index) and kidney dysfunction. ARIMA models forecasted ASPR and ASYR through 2036.

### Results

From 1990 to 2021, ASPR and ASYR increased globally and in all BRICS nations. In 2021, China recorded the highest ASPR and ASYR in males (1232.34 and 38.83 per 100,000, respectively), with EAPC of 1.14 and 1.13. Brazil had the lowest burden. In 2021, China and India reported the highest absolute burdens of gout, with approximately 16.79 million prevalent cases and 525,967 YLDs in China, and 5.32 million prevalent cases and 164,153 YLDs in India. High BMI and kidney dysfunction were key contributors to ASYR, especially in older adults. Forecasts indicate ASPR and ASYR will decline globally by 2036, but rise in Brazil, India, Russia, and South Africa. Chinese males show projected declines, while female rates increase modestly.

**Data availability statement:** The datasets analyzed during the current study are available in the http://ghdx.healthdata.org/gbd-results-tool.

**Funding:** This study was supported by Guangxi Clinical Research Center for Urology and Nephrology (grant No. Guike20297081). The corresponding author (Jiwen Cheng), who is the principal investigator of the grant, was involved in the study design, decision to publish, supervision and manuscript preparation.

**Competing interests:** The authors have declared that no competing interests exist.

## Conclusions

The burden of gout is rising across BRICS countries, shaped by aging populations, metabolic risk exposures, and distinct sex-specific trends. These findings highlight the importance of developing locally adapted prevention and control strategies to address this growing challenge.

## Introduction

Gout is the most prevalent inflammatory arthritis [1], marked by the deposition of monosodium urate (MUS) crystals in synovial joints and surrounding tissues, which triggers recurrent acute attacks, progressive joint damage, and chronic pain. The condition is tightly linked to hyperuricemia, arising from disrupted homeostasis between uric acid synthesis and renal or intestinal excretion [2]. Established risk factors of gout include male sex, aging, genetic susceptibility, obesity, excessive alcohol intake, purine-rich diets, and metabolic comorbidities such as fatty liver disease, arterial hypertension, chronic kidney disease (CKD), and insulin resistance [3]. Notably, impaired renal function not only exacerbates hyperuricemia but also contributes to the onset and progression of gout-related nephropathy [4], underscoring the clinical need for integrated management strategies targeting both systemic urate burden and kidney health.

Despite the availability of effective urate-lowering therapies, including xanthine oxidase inhibitors (such as allopurinol and febuxostat) and uricosuric agents (e.g., benzbromarone), the global management of gout remains suboptimal [1]. Many patients experience delayed diagnosis, poor adherence to long-term therapy, and limited access to appropriate medications, especially in low- and middle-income countries. Furthermore, gout is associated with significant healthcare costs, including hospitalizations, pharmacologic treatment, and productivity loss, representing a growing burden on patients and healthcare systems alike [5]. In recent decades, the global prevalence and disability burden of gout have increased steadily, driven by aging populations, nutritional transitions, and the rising prevalence of obesity and metabolic syndrome [6]. This trend is particularly evident in low- and middle-income regions undergoing rapid urbanization and lifestyle changes.

The BRICS nations—Brazil, Russia, India, China, and South Africa—together represent over 40% of the global population and exhibit diverse demographic and health system characteristics [7,8]. These countries are undergoing epidemiological transitions marked by a surge in non-communicable diseases (NCDs), including gout. Yet, comprehensive national assessments of gout burden within the BRICS context remain limited [9,10]. Most existing studies have focused on global or high-income settings, leaving critical gaps in understanding disease trends, risk factors, and healthcare disparities across emerging economies. The Global Burden of Disease (GBD) framework provides a robust foundation for generating standardized and comparable estimates across regions, time periods, and risk factors [11,12].

To date, no study has systematically examined gout burden trends across the BRICS nations using the most recent GBD data. This study seeks to fill that gap by analyzing GBD 2021 data to evaluate gout prevalence, years lived with disability (YLDs), and age-standardized rates (ASRs) from 1990 to 2021. It further assesses the contributions of two key metabolic risk factors—high body mass index (BMI) and impaired kidney function—to national disease burdens [13]. In addition, an autoregressive integrated moving average (ARIMA) model was applied to forecast 15-year trends (2022–2036) in age-standardized prevalence rate (ASPR) and YLD rate (ASYR), stratified by sex and country. By integrating historical trends and future projections, this study seeks to generate data-driven insights to support public health planning and targeted intervention strategies in emerging economies.

## Methods

### Study population and data collection

In this study, we utilized data from the GBD 2021 Study, obtained via the Global Health Data Exchange (https://ghdx.healthdata.org/gbd-2021). The database covers 371 diseases and injuries—including gout—and 88 risk factors, such as kidney dysfunction and BMI. The data encompass 204 countries and territories across 21 global regions, spanning the years 1990–2021 [11,13]. GBD data are derived from systematic reviews of published literature, government and international agency reports, primary sources such as Demographic and Health Surveys, and datasets contributed by GBD collaborators. Data processing involves standardized steps, including age–sex splitting, cause aggregation, and noise reduction, followed by modeling with tools such as the Cause of Death Ensemble model, spatiotemporal Gaussian process regression, and DisMod-MR. Methodological details are available online (http://www.healthdata.org/gbd/about/protocol) and have been described previously. This study adheres to the GATHER (Guidelines for Accurate and Transparent Health Estimates Reporting) statement, and all data used are de-identified and aggregated, as provided by Health Metrics and Evaluation (IHME) [14].

This study analyzed the age-, sex-, and location-specific prevalence, YLDs, and ASR of gout in BRICS countries, along with corresponding 95% uncertainty intervals (UI). YLDs quantify the non-fatal burden of disease by accounting for both the severity and duration of health loss. We also assessed YLDs attributable to specific risk factors. The joinpoint regression analysis (version 5.1.0) was employed to examine temporal trends in ASYR attributable to high BMI and kidney dysfunction among individuals with gout from 1990–2021. The average annual percentage change (AAPC) was calculated as the geometric mean of annual percentage change (APC) estimates. The methodology of GBD 2021 has been extensively described in previous publications [15,16]. As the study relied on publicly available, de-identified data, it was exempt from ethical review under Article 32 of the Declaration of Helsinki. Accordingly, informed consent was not required [9].

In GBD 2021, gout is classified under the International Classification of Disease 10th revision (ICD-10) code M10. Case identification follows the American College of Rheumatology's 1977 criteria, requiring MSU crystals in joint fluid, a tophus confirmed to contain MSU crystals, or the presence of at least 6 of 12 clinical features consistent with gout [6].

### Risk factor analysis

The gout-related disease burden attributable to two key metabolic risk factors—elevated BMI and kidney dysfunction—was analyzed. These represented the only level 2 risk factors in the GBD results tool with available ASYR. Given the established role of obesity in gout pathogenesis, BMI was considered a primary risk factor. BMI (kg/m²) serves as a standard anthropometric measure for classifying body size and identifying overweight or obesity. A high BMI was defined as ≥ 25 kg/m² for individuals aged ≥ 20 years [14]. Kidney dysfunction was also included, as it is defined within the GBD framework as a risk factor for gout. We acknowledge that in clinical settings gout and kidney dysfunction have a bidirectional association; however, within the GBD risk–outcome framework, kidney dysfunction is modeled only as a contributor to gout burden, and not as an outcome of gout. Details of data sources and selection procedures are available in prior publications [13,15].

## ARIMA forecasting

To forecast 15-year trends (2022–2036) in gout burden across BRICS countries, we applied the autoregressive integrated moving average (ARIMA) model to two age-standardized indicators: ASPR and ASYR, stratified by sex and country [17]. ARIMA is a time-series modeling technique that captures underlying temporal patterns through three parameters: *p* (autoregressive order), *d* (degree of differencing), and *q* (moving average order) [18]. It is particularly suited for long-term forecasting of disease trends, as it accommodates complex autocorrelations and non-stationarity in the data. ARIMA has been widely applied in epidemiological projections, including recent GBD-based forecasts of cancer incidence (e.g., thyroid cancer among adolescents and young adults from 2022 to 2050), as well as studies on non-cancer diseases, such as the projected burden of intracerebral hemorrhage in the Asian population aged 45 and older from 2022 to 2041 [18,19].

## Statistical analysis

The ASR (per 100,000 population) was calculated by summing the products of age-specific rates ($a_i$) and the corresponding weights ($w_i$) of the standard population for each age group i, then dividing by the total standard population. Subsequently, this sum was normalized by dividing it by the aggregate sum of the standard population weights: $ASR = \frac{\sum_{i=1}^{A} a_i w_i}{\sum_{i=1}^{A} w_i} \times 100,000$. Temporal trends in ASRs over time were assessed using estimated annual percentage changes (EAPC). Assuming a linear relationship between the natural logarithm of ASR and calendar year, EAPC were derived from the regression model: ln (ASR) = α + βx + ε, where x denotes year and ε is the error term. EAPC was computed as 100 × (exp(β) − 1), with 95% confidence intervals (CI) estimated from the model. An increasing trend was indicated if both the EAPC and its lower 95% CI exceeded zero; a decreasing trend was inferred if both the EAPC and upper 95% CI were below zero. All statistical analyses and visualizations were performed using R software (version 4.1.2), with $P < 0.05$ considered statistically significant.

## Results

### Trends in global and BRICS countries gout burden, 1990–2021

Fig 1 illustrates the trends in the ASPR and ASYR of gout among males and females from 1990 to 2021 at both global and BRICS country levels. Overall, both ASPR and ASYR demonstrated an upward trend over time in males and females across the globe and in all five BRICS nations. Notably, China exhibited substantially higher ASPR and ASYR values in both sexes compared to the global average and the other four BRICS countries, with the most pronounced increases over the study period. In contrast, the other four BRICS countries generally showed lower ASPR and ASYR than the global levels, with Brazil reporting the lowest burden among them. Additionally, in all regions and time points, males consistently had higher ASPR and ASYR values than females.

Table 1 presents the ASPR and ASYR estimates in 2021 and the corresponding EAPCs from 1990 to 2021. Globally and across all BRICS countries, the EAPCs for ASPR and ASYR were consistently positive, indicating a growing burden of gout. In 2021, the global ASPR of gout was 653.82 (95% UI: 526.13–810.46) per 100,000 population for both sexes, with 1021.84 (819.96–1264.60) in males and 315.05 (253.27–392.24) in females. The global ASYR was 20.22 (13.77–28.77) per 100,000 population in both sexes, 31.68 (21.50–45.13) in males, and 9.61 (6.53–13.95) in females. Among the BRICS nations, Chinese males had the highest ASPR in 2021 at 1232.34 (983.05–1532.59) per 100,000 population, along with the greatest EAPC for ASPR, at 1.14 (95% CI: 1.01–1.27). Similarly, Chinese males also exhibited the highest ASYR at 38.83 (26.07–55.49) per 100,000 population and the highest EAPC for ASYR, at 1.13 (95% CI: 1.00–1.26). In contrast, India exhibited the lowest EAPC for ASPR and ASYR in the overall (both-sex) population among BRICS countries, with values of 0.22 (95% CI: 0.19–0.24) and 0.24 (95% CI: 0.22–0.26), respectively.

As shown in Table 2, the absolute number of prevalent gout cases among Brazilian females increased from 71,759 (95% UI: 57,241–91,273) in 1990–241,169 (95% UI: 191,165–305,060) in 2021, representing the largest percentage

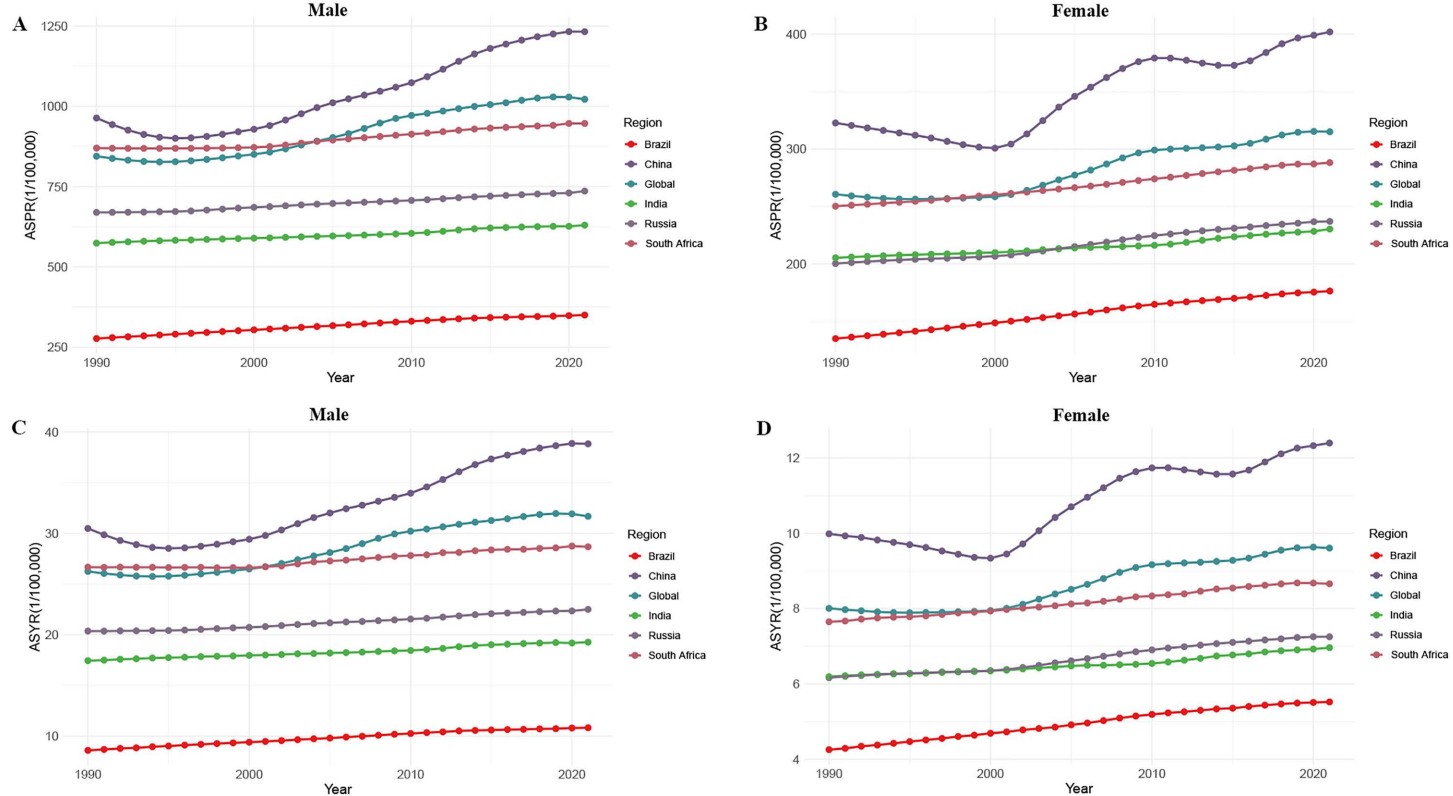

**Fig 1. Temporal trends in ASPR and ASYR of gout in BRICS (Brazil, Russia, India, China, and South Africa) countries and globally, 1990–2021.**
**A.** ASPR in males; **B.** ASPR in females; **C.** ASYR in males; **D.** ASYR in females. ASPR, age-standardized prevalence rates; ASYR, age-standardized years lived with disability rates.

increase (236.08%) among all BRICS countries. Likewise, the number of YLDs cases rose by 228.63% over the same period. In contrast, Russia showed the smallest percentage increases in both prevalence and YLDs cases during this timeframe.

## Age and sex distribution of gout burden in 1990 and 2021

As shown in Fig 2, from 1990 to 2021, the number of gout prevalence cases increased across almost all age groups for both males and females in the five BRICS countries. In Brazil, the age group with the highest number of prevalent gout cases in 1990 was 50–54 years for both sexes, which shifted to 55–59 years by 2021 (Fig 2A). In Russia, the peak age group in 1990 was 60–64 years for both males and females; by 2021, it remained 60–64 years for males but shifted to 65–69 years for females (Fig 2B). In India, the highest case numbers in 1990 occurred in the 55–59 age group for males and 60–64 for females; in 2021, the 60–64 age group had the highest number of prevalent cases for both sexes (Fig 2C). In China, the peak age group in 1990 was 55–59 years for both sexes, while in 2021, it remained 55–59 for males but shifted to 65–69 for females (Fig 2D). In South Africa, the peak age group remained consistent at 60–64 years for both males and females in both 1990 and 2021 (Fig 2E). Overall, in both 1990 and 2021, males consistently had a greater number of prevalent gout cases than females within the same age groups across all five BRICS countries. In addition to these peak age groups, a broader pattern was observed across countries. In all BRICS nations, the gout burden was concentrated among middle-aged and older adults, most prominently within the 50–69 year age range, with a gradual shift toward older age groups from 1990 to 2021. This shift was particularly evident in China and Russia.

**Table 1. Trends in ASPR and ASYR of gout in BRICS (Brazil, Russia, India, China, and South Africa) countries and globally, 1990–2021.**

| Country | Year | Sex | ASPR (per 100,000) (95% UI) | EAPC of ASPR (95% CI) | ASYR (per 100,000) (95% UI) | EAPC of ASYR (95% CI) |
|---|---|---|---|---|---|---|
| Brazil | 1990 | Both | 201.02 (161.4–251.02) | – | 6.29 (4.18–9.09) | – |
| | | Male | 277.05 (223.46–343.92) | – | 8.61 (5.82–12.36) | – |
| | | Female | 134.84 (107.29-172.78) | – | 4.25 (2.75–6.27) | – |
| | 2021 | Both | 255.36 (205.69–318.19) | 0.81 (0.78–0.84) | 7.95 (5.28–11.32) | 0.80 (0.77–0.83) |
| | | Male | 350.21 (283.12–433.98) | 0.78 (0.75–0.82) | 10.83 (7.29–15.39) | 0.78 (0.74–0.81) |
| | | Female | 176.23 (139.98–222.79) | 0.91 (0.88–0.94) | 5.52 (3.58–7.97) | 0.89 (0.86–0.93) |
| Russia | 1990 | Both | 372.33 (297.53–465.69) | – | 11.45 (7.62–16.39) | – |
| | | Male | 669.64 (540.42–834.91) | – | 20.36 (13.64–28.94) | – |
| | | Female | 200.2 (160.39–252.75) | – | 6.16 (4.11–8.86) | – |
| | 2021 | Both | 439.04 (349.75–548.14) | 0.55 (0.54–0.56) | 13.48 (9.01–19.43) | 0.56 (0.54–0.58) |
| | | Male | 736.06 (590.99–917.36) | 0.32 (0.31–0.33) | 22.50 (15.15–32.41) | 0.36 (0.34–0.37) |
| | | Female | 236.9 (189.05–296.51) | 0.61 (0.58–0.64) | 7.25 (4.85–10.58) | 0.60 (0.56–0.63) |
| India | 1990 | Both | 394.93 (317.02–493.19) | – | 11.99 (8.05–17.28) | – |
| | | Male | 574.05 (462.13–713.85) | – | 17.44 (11.69–25.09) | – |
| | | Female | 205.2 (164.08–258.46) | – | 6.19 (4.14–9.04) | – |
| | 2021 | Both | 424.26 (340.6–530.02) | 0.22 (0.19–0.24) | 12.96 (8.67–18.71) | 0.24 (0.22–0.26) |
| | | Male | 629.95 (506.96–784.82) | 0.30 (0.29–0.32) | 19.27 (12.91–27.89) | 0.33 (0.31–0.35) |
| | | Female | 230.24 (184.38–288.6) | 0.36 (0.33–0.39) | 6.96 (4.70–9.90) | 0.37 (0.35–0.40) |
| China | 1990 | Both | 640.68 (512.12–796.74) | – | 20.2 (13.46–29.14) | – |
| | | Male | 963.57 (771.22–1193.56) | – | 30.48 (20.33–43.76) | – |
| | | Female | 322.72 (258.7–403.63) | – | 9.99 (6.68–14.70) | – |
| | 2021 | Both | 810.36 (644.76–1009.05) | 1.08 (0.95–1.20) | 25.43 (17.16–36.31) | 1.06 (0.94–1.19) |
| | | Male | 1232.34 (983.05–1532.59) | 1.14 (1.01–1.27) | 38.83 (26.07–55.49) | 1.13 (1.00–1.26) |
| | | Female | 401.98 (321.46–506.57) | 0.97 (0.81–1.13) | 12.40 (8.34–18.09) | 0.96 (0.79–1.12) |
| South Africa | 1990 | Both | 516.9 (413.66–641.63) | – | 15.87 (10.79–23.08) | – |
| | | Male | 869.99 (692.46–1081.43) | – | 26.67 (18.15–38.97) | – |
| | | Female | 250.18 (201.94–314.14) | – | 7.65 (5.21–11.18) | – |
| | 2021 | Both | 568.28 (454.94–709.03) | 0.37 (0.32–0.41) | 17.23 (11.7–25.06) | 0.33 (0.29–0.37) |
| | | Male | 946.68 (757.62–1187.78) | 0.33 (0.30–0.35) | 28.68 (19.66–42.07) | 0.29 (0.26–0.32) |
| | | Female | 288.17 (230.85–360.45) | 0.49 (0.48–0.50) | 8.66 (5.83–12.49) | 0.44 (0.43–0.46) |
| Global | 1990 | Both | 536.54 (430.28–665.72) | – | 16.67 (11.25–23.95) | – |
| | | Male | 844.63 (678.4–1049.92) | – | 26.25 (17.72–37.84) | – |
| | | Female | 260.55 (209.31–324.61) | – | 8.01 (5.41–11.66) | – |
| | 2021 | Both | 653.82 (526.13–810.46) | 0.87 (0.80–0.95) | 20.22 (13.77–28.77) | 0.86 (0.78–0.93) |
| | | Male | 1021.84 (819.96–1264.60) | 0.87 (0.79–0.94) | 31.68 (21.5–45.13) | 0.86 (0.78–0.93) |
| | | Female | 315.05 (253.27–392.24) | 0.81 (0.73–0.90) | 9.61 (6.53–13.95) | 0.79 (0.71–0.87) |

ASPR, age-standardized prevalence rates; ASYR, age-standardized years lived with disability rates; EAPC, estimated annual percentage change; CI, confidence interval; UI, uncertainty interval.

## Temporal tends in gout ASYR attributable to high BMI and kidney dysfunction by sex

Fig 3 describes the sex-stratified joinpoint analysis of gout ASYR attributable to high BMI and kidney dysfunction in BRICS countries from 1990 to 2021. Overall, an increasing trend in ASYR attributable to high BMI was observed in five BRICS countries, particularly among males. In China, the most pronounced increases in ASYR attributable to high BMI among males

**Table 2. Changes in gout prevalent and YLDs cases in BRICS (Brazil, Russia, India, China, and South Africa) countries and globally, 1990–2021.**

| Country | Sex | Prevalent Cases 95% UI 1990 | Prevalent Cases 95% UI 2021 | Change (%) | YLDs Cases 95% UI 1990 | YLDs Cases 95% UI 2021 | Change (%) |
|---|---|---|---|---|---|---|---|
| Brazil | Both | 203,435 (161,477–253,747 | 648,012 (519,467–814,679) | 219.61 | 6,474 (4,229–9,338) | 20,204 (13,437–28,951) | 212.08 |
| | Male | 131,676 (104,664–163,665) | 406,843 (326,666–509,639) | 208.97 | 4,180 (2,769–6,049) | 12,663 (8,530–18,155) | 202.94 |
| | Female | 71,759 (57,241–91,273) | 241,169 (191,165–305,060) | 236.08 | 2,295 (1,468–3,360) | 7,542 (4,910–10,884) | 228.63 |
| Russia | Both | 661,947 (521,024–836,506) | 987,087 (775,580–1,246,321) | 49.12 | 20,366 (13,575–28,855) | 30,076 (20,345–43,579) | 47.68 |
| | Male | 438,094 (344,185–557,336) | 658,050 (514,931–836,310) | 50.21 | 13,513 (8,996–19,473) | 20,162 (13,613–29,318) | 49.20 |
| | Female | 223,853 (177,887–285,447) | 329,037 (260,288–414,069) | 46.99 | 6,852 (4,581–9,938) | 9,914 (6,735–14,610) | 44.69 |
| India | Both | 2,012,110 (1,591,668–2,549,676) | 5,315,284 (4,217,470–6,672,708) | 164.16 | 62,529 (41,624–90,358) | 164,153 (109,106–232,910) | 162.52 |
| | Male | 1,515,642 (1,195,418–1,921,933) | 3,856,499 (3,071,064–4,841,750) | 154.45 | 47,163 (31,148–68,590) | 119,598 (79,450–169,702) | 153.58 |
| | Female | 496,468 (394,518–631,568) | 1,458,786 (1,161,534–1,843,354) | 193.83 | 15,366 (10,095–22,043) | 44,555 (29,798–63,435) | 189.96 |
| China | Both | 5,971,856 (4,746,981–7,526,175) | 16,788,147 (13,144,120–21,277,610) | 181.12 | 190,614 (126,028–276,353) | 525,967 (353,075–758,229) | 175.93 |
| | Male | 4,534,505 (3,573,984–5,733,061) | 12,484,534 (9,811,161–15,833,884) | 175.32 | 145,587 (96,540–213,926) | 393,759 (263,612–569,208) | 170.46 |
| | Female | 1,437,350 (1,139,577–1,824,401) | 4,303,613 (3,369,653–5,485,748) | 199.41 | 45,028 (30,017–65,809) | 132,209 (87,761–193,962) | 193.62 |
| South Africa | Both | 115,000 (91,679–144,253) | 282,940 (223,685–357,508) | 146.03 | 3,576 (2,424–5,196) | 8,673 (5,830–12,616) | 142.53 |
| | Male | 84,211 (67,335–105,376) | 203,682 (161,951–258,623) | 141.87 | 2,625 (1,766–3,823) | 6,274 (4,241–9,175) | 139.01 |
| | Female | 30,789 (24,786–38,701) | 79,259 (62,975–99,983) | 157.42 | 951 (643–1,383) | 2,399 (1,602–3,486) | 152.26 |
| Global | Both | 22,264,515 (17,793,190–27,965,605) | 56,474,573 (45,161,987–70,288,316) | 153.65 | 697,541 (470,078–1,000,259) | 1,747,546 (1,186,175–2,484,547) | 150.53 |
| | Male | 16,663,577 (13,195,760–20,967,927) | 42,129,942 (33,505,355–52,848,503) | 152.83 | 524,492 (351,784–758,449) | 1,311,275 (886,339–1,879,934) | 150.00 |
| | Female | 5,600,938 (4,492,022–7,003,055) | 14,344,631 (11,461,922–17,901,240) | 156.11 | 173,048 (116,482–253,077) | 436,272 (296,522–635,341) | 152.11 |

YLDs,Years lived with disability; UI,uncertainty interval

were observed during 2001–2004 and 2010–2015, with APCs exceeding 3.89%. In India, a notable increase was observed among males during 2001–2004, with an APC of 3.11%. In Russia and South Africa, ASYR attributable to high BMI showed relatively stable upward trends among both males and females over the past 32 years (Fig 3A–3C). For ASYR attributable to kidney dysfunction, distinct temporal patterns were observed. In China, both males and females experienced marked declines during 1990–1994, with APCs of –5.48% and –5.69%, respectively. However, from 2000 to 2005, the APC for Chinese females increased significantly to 3.52%. In India, ASYR attributable to impaired kidney function showed a downward trend among both sexes over the 32-year period. In contrast, Brazil and South Africa exhibited a relatively steady upward trend in ASYR attributable to renal impairment among both males and females (Fig 3D–3F).

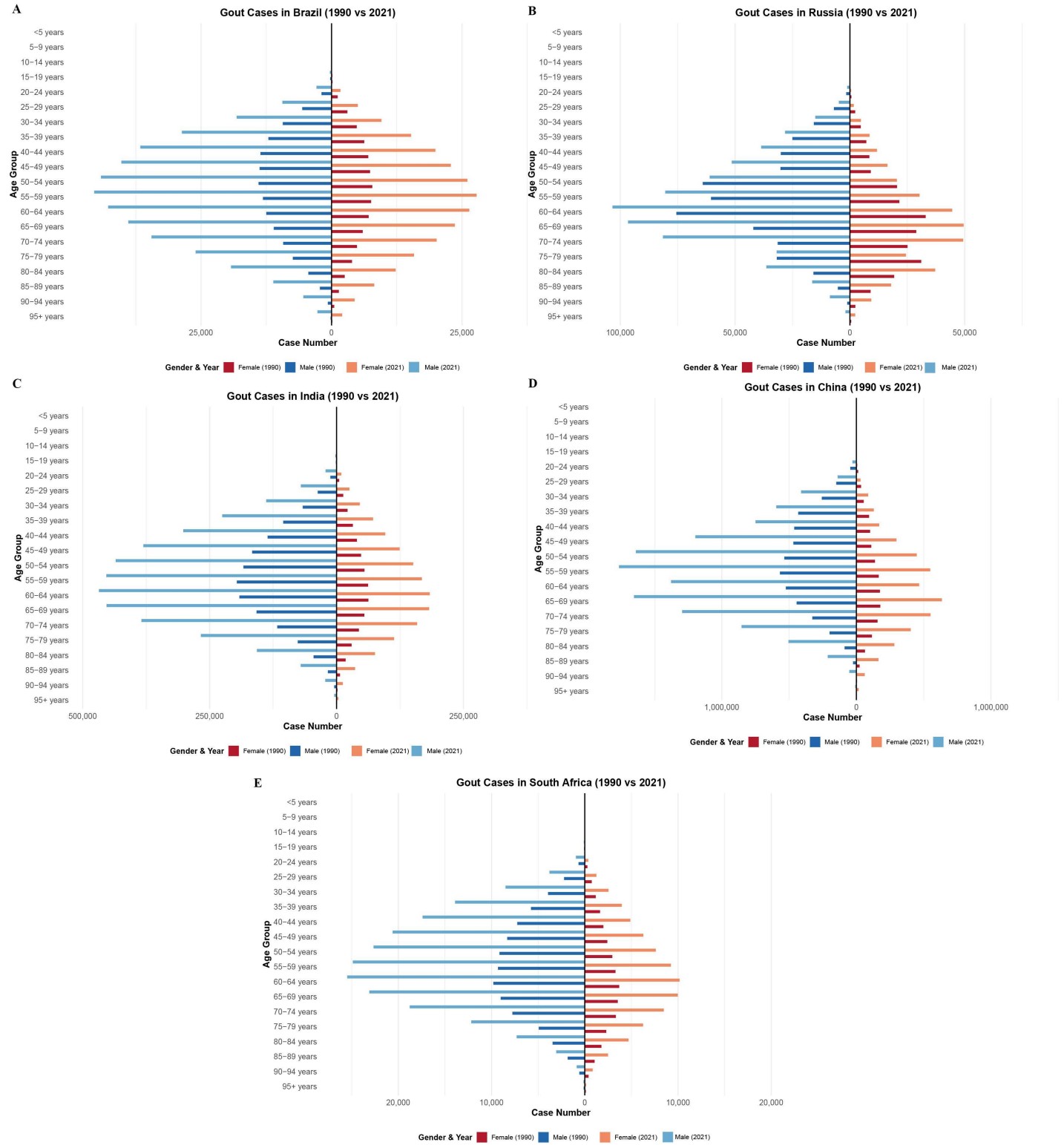

**Fig 2. Age-specific number of gout prevalent cases by sex in BRICS (Brazil, Russia, India, China, and South Africa) nations, 1990 and 2021.**
A-E: Brazil, Russia, India, China, and South Africa, respectively.

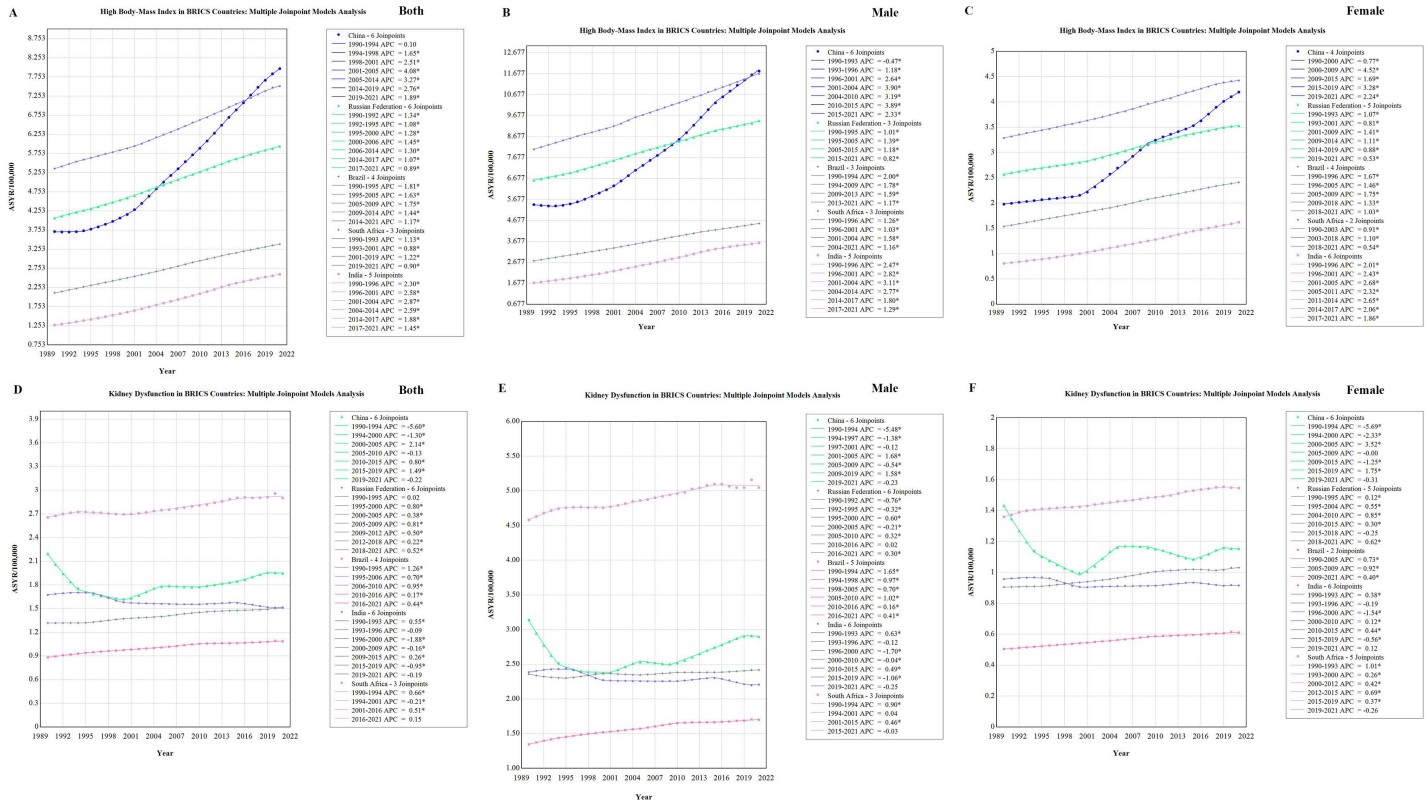

**Fig 3. Joinpoint regression analysis of ASYR of gout attributable to high BMI and kidney dysfunction in BRICS (Brazil, Russia, India, China, South Africa) countries, 1990–2021.** ASYR trends attributable to high BMI: A. both sexes, B. males, C. females. ASYR trends attributable to kidney dysfunction: D. both sexes, E. males, F. females. *Indicates a statistically significant annual percent change (APC) at *P* < 0.05. ASYR, age-standardized years lived with disability rates; BMI: body mass index.

### Age-specific YLDs attributable to high BMI in BRICS nations

Fig 4 displays age-specific YLDs rates for gout attributable to high BMI in the BRICS countries in 1990 and 2021. In most countries, the burden was greatest among older adults (typically aged ≥55 years) and rose with age. From 1990 to 2021, YLDs increased markedly among middle-aged and elderly populations, particularly in China and India. In contrast, increases were more modest in Brazil, Russia, and South Africa. In both years, the highest YLDs rates were consistently observed in the 95 + age group. Russia and South Africa reported higher YLDs across all age brackets and both sexes than the other three BRICS nations. Fig 5 shows comparable age-specific patterns in YLDs due to kidney dysfunction. These also increased with age in both sexes across all five countries, especially after age 65. South Africa showed the steepest increases over time. Compared to 1990, Russia experienced the smallest rise in 2021, while South Africa exhibited the most pronounced surge among individuals over 60.

### Projected trends in gout ASPR and ASYR, 2022–2036: global and BRICS perspectives

Figs 6 and 7 present ARIMA-based projections of ASPR and ASYR for gout from 2022 to 2036 across global and BRICS populations. Globally, both indicators are expected to decline steadily. By 2036, the ASPR is projected to reach 893.57 per 100,000 in males and 307.93 in females. In China, ASPR among males is anticipated to fall to 966.35, while female rates may increase to 424.24. Meanwhile, Brazil, Russia, India, and South Africa are forecasted to see rising ASPR trends, most notably among Russian men and Indian women (Fig 6E, 6H). For ASYR, a gradual global decline is also

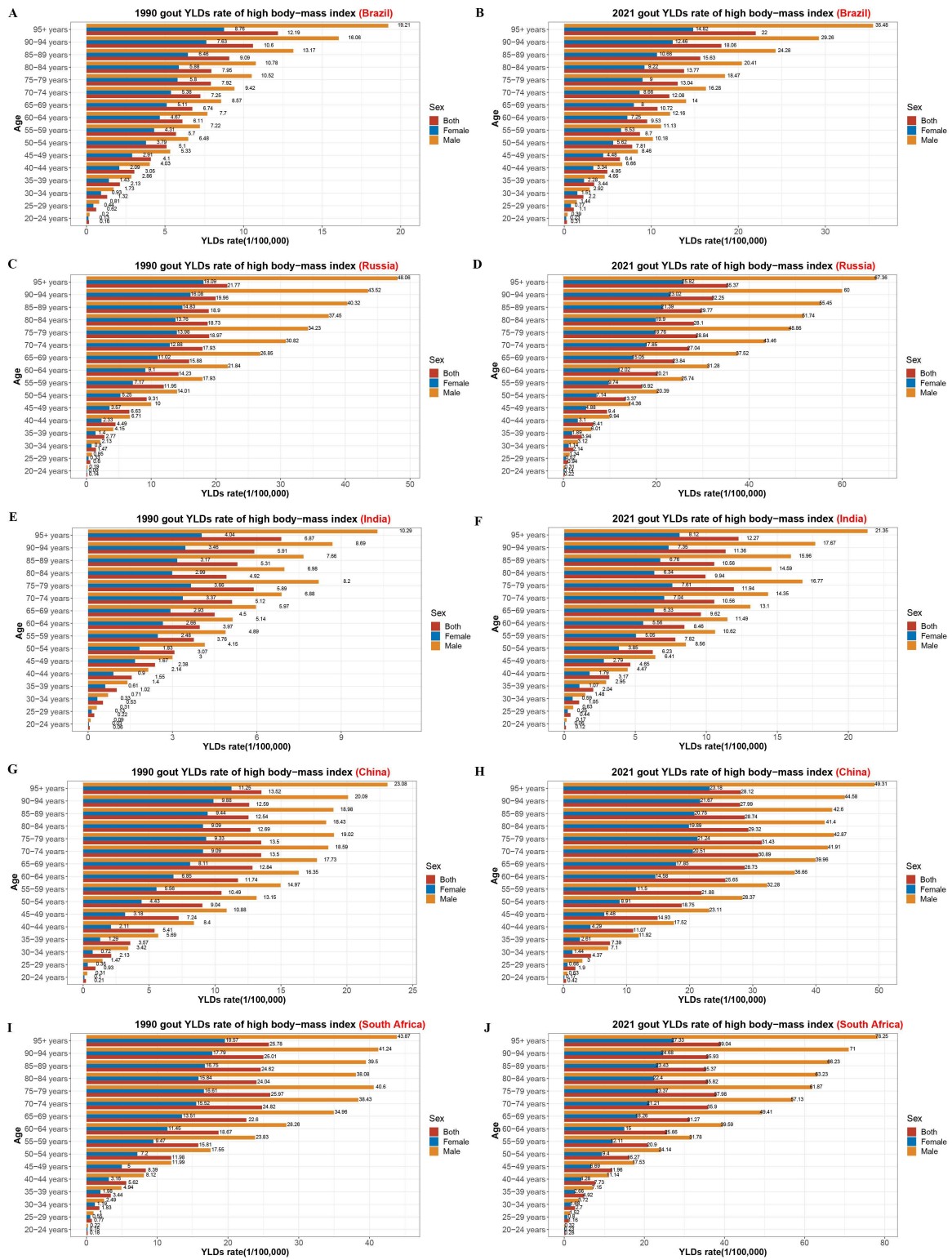

**Fig 4. Age-specific YLD rates of gout attributable to high BMI by sex in BRICS (Brazil, Russia, India, China, South Africa) nations in 1990 and 2021.** A–B: Brazil (1990, 2021); C–D: Russia (1990, 2021); E–F: India (1990, 2021); G–H: China (1990, 2021); I–J: South Africa (1990, 2021). YLD, years lived with disability; BMI: body mass index.

**Fig 5. Age-specific YLD rates of gout attributable to kidney dysfunction by sex in BRICS (Brazil, Russia, India, China, South Africa) nations in 1990 and 2021.** A–B: Brazil (1990, 2021); C–D: Russia (1990, 2021); E–F: India (1990, 2021); G–H: China (1990, 2021); I–J: South Africa (1990, 2021). YLD, years lived with disability.

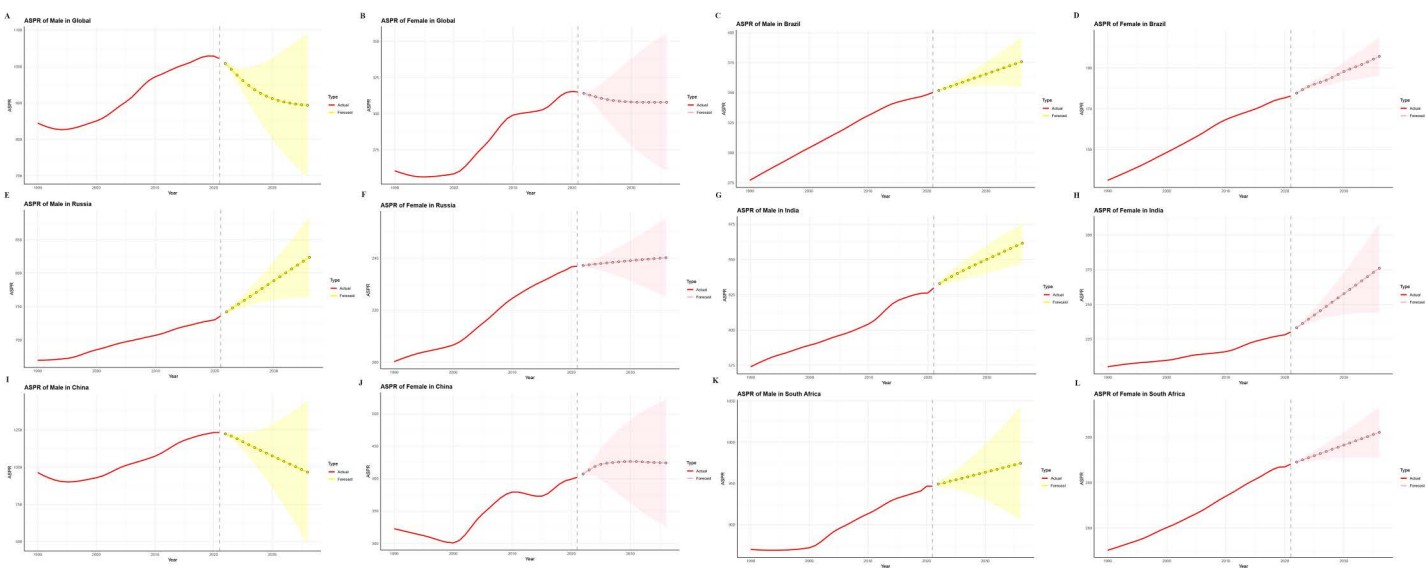

**Fig 6. Temporal trends (1990–2021) and ARIMA-based forecasts (2022–2036) of gout ASPR in BRICS (Brazil, Russia, India, China, South Africa) nations and globally.** A–B: Global ASPR in males and females; C–D: Brazil; E–F: Russia; G–H: India; I–J: China; K–L: South Africa. ASPR, age-standardized prevalence rates; ARIMA, Autoregressive integrated moving average.

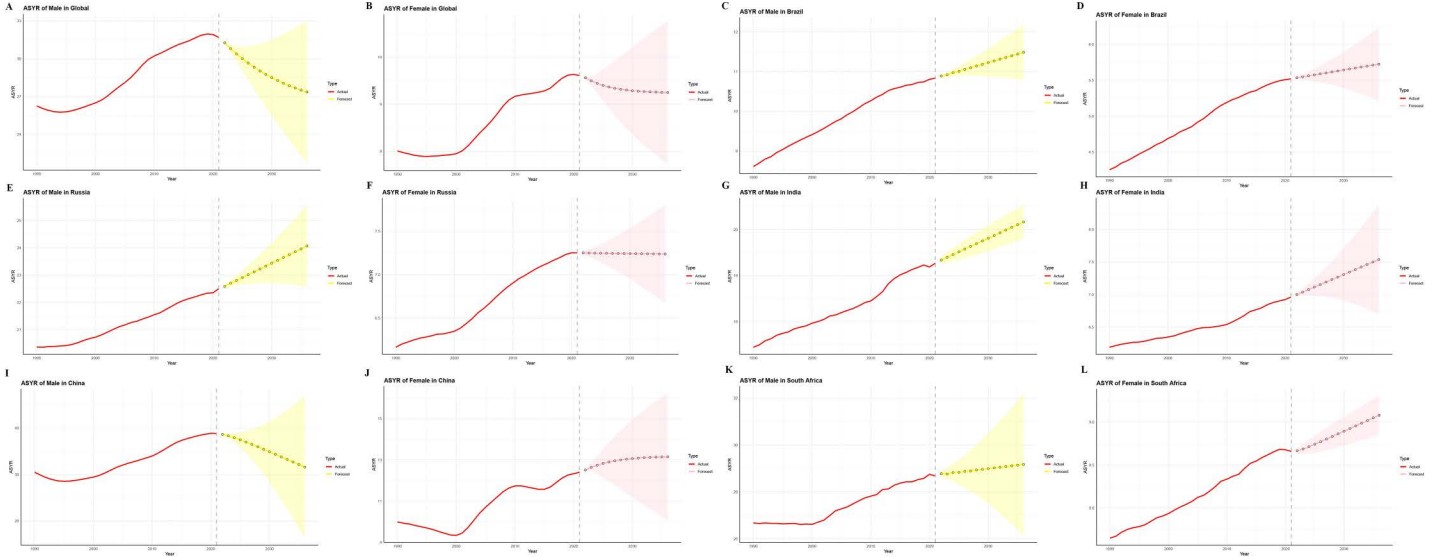

**Fig 7. Temporal trends (1990–2021) and ARIMA-based forecasts (2022–2036) of gout ASYR in BRICS (Brazil, Russia, India, China, South Africa) nations and globally.** A–B: Global ASPR in males and females; C–D: Brazil; E–F: Russia; G–H: India; I–J: China; K–L: South Africa. ASYR, age-standardized years lived with disability rates; ARIMA, Autoregressive integrated moving average.

projected—falling to 27.35 in males and 9.25 in females by 2036. In China, male ASYR is expected to decrease to 31.59, while female ASYR may rise to 13.15, mirroring the ASPR trend. In Russia, male ASYR is projected to increase, while female rates are expected to remain largely unchanged (Fig 7E–7F). For the other BRICS nations, projected ASYR trajectories align closely with those of ASPR.

## Discussion

To our knowledge, this is the first study to provide a detailed, cross-country comparison of gout burden in BRICS nations using the most recent GBD 2021 data. Between 1990 and 2021, both the ASPR and ASYR associated with gout showed a consistent upward trend globally and across all five countries. China recorded the highest burden, particularly among men, where the EAPC reached 1.14 for ASPR and 1.13 for ASYR. These increases likely reflect a combination of rapid population aging, increasing urbanization, evolving dietary habits, and a growing prevalence of metabolic disorders such as obesity, hypertension, and CKD [1,20]. Notably, while China bore the highest age-standardized rates, India and China together accounted for the majority of absolute gout burden. In 2021 alone, China recorded approximately 16.79 million prevalent cases and 5.26 million YLDs cases, far exceeding other BRICS countries. These absolute figures, driven by massive population bases, underscore the need to consider both relative disease rates and absolute case numbers in health system planning. While ASR facilitate comparisons across countries and time, absolute numbers reflect tangible healthcare demands and inform infrastructure and workforce capacity planning [7]. Sex differences in gout burden were observed consistently, with men showing markedly higher ASPR and ASYR than women. However, projections suggest a narrowing of this gap in some regions. For instance, male rates in China are expected to decline by 2036, whereas female rates may rise—possibly due to increasing obesity among older women and postmenopausal changes in uric acid metabolism. These trends underscore the need for sex-specific prevention and treatment strategies, particularly in aging populations [21].

In line with prior global analyses, gout-related disability in BRICS countries was predominantly observed in older age groups, especially individuals aged ≥ 55 years [9]. The observed concentration of gout burden within the 50–69 year age range across BRICS countries reflects the well-established impact of aging and metabolic comorbidities on gout risk. The gradual shift toward older age groups in recent decades may further highlight the influence of population aging and warrants targeted management strategies for middle-aged and elderly populations. Between 1990 and 2021, the YLDs rate attributable to high BMI increased substantially in this demographic, most notably in China, India, and South Africa. The findings highlight the synergistic effects of aging and metabolic risk accumulation in driving non-fatal gout burden. In contrast, Russia and Brazil displayed more modest increases, potentially reflecting slower demographic transitions or differential healthcare access. Kidney dysfunction, another major contributor to gout burden, demonstrated clear age-dependent trends [4]. In all BRICS countries, YLDs rates rose sharply after age 65, peaking in the oldest age groups. South Africa consistently showed the highest age-specific ASYR attributable to kidney dysfunction, which may relate to a high prevalence of hypertension, diabetes, and HIV-related renal damage, as noted in prior studies [22,23]. Importantly, Russia exhibited the smallest increase in this measure over time, whereas South Africa experienced the largest, highlighting wide inter-country variation in metabolic and renal health profiles.

Joinpoint regression further revealed critical inflection periods in ASYR trends. For example, in China and India, male ASYR attributable to high BMI increased most steeply during 2001–2004 and 2010–2015, with APCs exceeding 3.1%. Similarly, ASYR due to kidney dysfunction declined in China during the early 1990s, followed by a significant rise in the 2000s—particularly among women. These fluctuations may reflect the lagged effects of shifting lifestyle patterns, diagnostic capacity, and health system responses. Identifying such time windows is crucial for aligning interventions with epidemiological transitions [24].

Despite heterogeneous trends across the five BRICS nations, both the relative and absolute burdens of gout have increased markedly from 1990 to 2021. This escalation may partially reflect population growth and demographic aging, but rising diagnostic rates and improved disease awareness could also contribute to the observed increases in prevalence, ASPR [25]. The growing ASYR among older adults deserves particular attention. Gout in this age group often co-occurs with other chronic illnesses, intensifying the risk of disability and reducing quality of life. Early identification and integrated management of gout and its comorbidities could help limit long-term disability and reduce healthcare demands. Perhaps more concerning is the upward trend in gout cases among adults aged 20–39, suggesting the disease may be affecting

individuals earlier in life than previously recognized [26]. This shift is particularly troubling given the potential for long-term complications if untreated. Coordinated efforts across BRICS countries—ranging from targeted screening programs to lifestyle-based interventions and patient education—are urgently needed to reverse this trend and reduce the growing impact of gout across age groups.

Looking forward, our ARIMA-based projections suggest a modest global decline in both ASPR and ASYR of gout by 2036, likely reflecting better awareness, improved access to urate-lowering therapies, and enhanced management of metabolic comorbidities [27,28].However, these projections should be interpreted with caution, as ARIMA models are inherently based on historical data and may not fully capture future changes driven by evolving health policies, socioeconomic transitions, or structural shifts in healthcare systems—factors that are particularly dynamic in BRICS countries. Moreover, these projected trends are not uniform across BRICS countries. While China is expected to see decreasing rates in males, the burden in females is projected to increase. In contrast, Brazil, India, and South Africa are likely to face continued growth in both ASPR and ASYR across sexes. Russia presents a more complex picture, with rising male burden and relatively stable female rates.

These country- and sex-specific forecasts have direct policy implications, highlighting the urgency of context-sensitive responses tailored to each nation's unique epidemiological and healthcare realities [29]. In India, where the gout burden is projected to rise across both sexes, national programs should prioritize strengthening primary healthcare infrastructure to support early metabolic screening and gout diagnosis, especially in rural and peri-urban regions. Investment in health worker training and affordable access to urate-lowering therapies could mitigate progression to chronic gout and its complications. In South Africa, addressing structural barriers—such as fragmented care delivery and limited specialist access—will be essential. Culturally adapted public health campaigns to raise awareness of dietary triggers, along with community-based interventions targeting obesity and hypertension, may yield long-term benefits, particularly among disadvantaged female populations. In contrast, China's growing burden in postmenopausal women underscores the need for integrated management of obesity, CKD, and hyperuricemia within women's health services [30]. Importantly, these projections emphasize the value of integrating gout into broader NCDs prevention frameworks, especially in BRICS countries where cardiovascular disease, diabetes, and dyslipidemia continue to rise. National health plans should incorporate salt and purine reduction policies, glucose and lipid management, and routine CKD screening, particularly for older adults and high-risk groups [3,9]. Multisectoral approaches—including nutrition education, primary care strengthening, and urban planning to promote active lifestyles—are also needed to curb the growing metabolic burden.

This analysis draws upon the most recent GBD 2021 dataset, enabling standardized cross-national comparisons across five large and diverse economies. By integrating time-trend analysis, risk attribution, and forecasting models, the study provides a broad and policy-relevant perspective on gout burden. Nevertheless, several limitations warrant consideration. First, GBD estimates are modeled from multiple sources and may be influenced by underreporting or inconsistent data quality—particularly in low- and middle-income regions. Second, differences in diagnostic practices, healthcare access, and clinical awareness across countries could introduce variability in disease ascertainment. Third, our projections assume a continuation of past trends and may not fully account for disruptive changes such as new treatments, healthcare reforms, or economic shocks. Fourth, although the GBD framework attributes gout burden to high BMI and impaired kidney function, it does not currently incorporate other well-established risk factors such as purine-rich diets, alcohol consumption, or genetic predispositions. This may lead to an underestimation of the role of lifestyle and behavioral determinants in shaping national and regional trends.

To improve the precision and utility of gout surveillance in BRICS countries, future research should aim to refine the scope of risk factor modeling. Including additional exposures—such as purine-rich diets, alcohol intake, and genetic variants—could better capture the full spectrum of disease risk [8]. Moreover, longitudinal cohort studies are needed to explore the mechanistic links between metabolic disorders, renal function, and urate accumulation, especially in aging and female populations. From a policy standpoint, country-specific evaluations of cost-effectiveness for screening, early

intervention, and urate-lowering therapies could help optimize resource allocation in constrained healthcare settings [5]. Given the frequent overlap of gout with other non-communicable diseases, cross-disciplinary collaboration between rheumatologists, nephrologists, cardiologists, and public health professionals will be essential for developing integrated, patient-centered care models [31]. Ultimately, translating burden data into targeted, equitable interventions remains a central challenge for reducing the long-term impact of gout in emerging economies.

## Conclusion

Over the past three decades, the burden of gout has increased across BRICS nations and globally, with China and India reporting the highest absolute cases. Age-standardized burden rose in all five countries, with notable sex- and region-specific patterns. ARIMA forecasts indicate further increases through 2036, especially in India, Russia, South Africa, Brazil, and Chinese females. These trends may reflect aging populations and rising metabolic risks. Targeted prevention strategies—including early detection, risk factor control, and equitable treatment access—are urgently needed to mitigate the growing gout burden in BRICS countries undergoing rapid demographic and health system transitions.

## Acknowledgments

The authors appreciate the works by the Global Burden of Disease Study 2021 collaborators.

## Author contributions

**Conceptualization:** Qizhou Mo, Shufeng Luo, Fengyi Wang, Min Qin, jiwen Cheng.

**Data curation:** Qizhou Mo, Shufeng Luo, Fengyi Wang, Haiqi Liang, Jiayin Yu, Min Qin.

**Formal analysis:** Qizhou Mo, Shufeng Luo, Fengyi Wang, Jiayin Yu, Min Qin.

**Funding acquisition:** jiwen Cheng.

**Investigation:** Qizhou Mo, Fengyi Wang, Haiqi Liang, Naikai Liao.

**Methodology:** Qizhou Mo, Fengyi Wang, Haiqi Liang, Jiayin Yu, Naikai Liao.

**Resources:** jiwen Cheng.

**Software:** Qizhou Mo, Shufeng Luo, Fengyi Wang, Haiqi Liang, Jiayin Yu.

**Supervision:** Qizhou Mo, Min Qin, jiwen Cheng.

**Validation:** Shufeng Luo, Fengyi Wang.

**Visualization:** Qizhou Mo, Shufeng Luo, Fengyi Wang, Min Qin.

**Writing – original draft:** Qizhou Mo, Shufeng Luo, Fengyi Wang, Haiqi Liang, Jiayin Yu, Naikai Liao, Min Qin, jiwen Cheng.

**Writing – review & editing:** Qizhou Mo, Min Qin, jiwen Cheng.

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
