## [Decision Letter · Decision Letter 0]

14 Jul 2025

Dear Dr. Cheng,

Thank you for submitting your manuscript to PLOS ONE. After careful consideration, we feel that it has merit but does not fully meet PLOS ONE’s publication criteria as it currently stands. Therefore, we invite you to submit a revised version of the manuscript that addresses the points raised during the review process.

We look forward to receiving your revised manuscript.

Kind regards,

Junzheng Yang

Academic Editor

PLOS ONE

Journal Requirements: 

 [This study was supported by Guangxi Clinical Research Center for Urology and Nephrology (grant No. Guike20297081).]. 

Reviewers' comments:

Reviewer's Responses to Questions

**Comments to the Author**

1. Is the manuscript technically sound, and do the data support the conclusions?

Reviewer #1: Yes

Reviewer #2: Yes

2. Has the statistical analysis been performed appropriately and rigorously?

Reviewer #1: Yes

Reviewer #2: Yes

3. Have the authors made all data underlying the findings in their manuscript fully available?

Reviewer #1: Yes

Reviewer #2: Yes

4. Is the manuscript presented in an intelligible fashion and written in standard English?

Reviewer #1: Yes

Reviewer #2: Yes

Reviewer #1: This manuscript addresses a relevant subject such as gout and a prevalent condition such as obesity. On behalf of the metodological approach, it is consistent and have been adequately explained. Even though the gap of years from the original data and the analysis exceeds 3 years, the forecast ARIMA is an appropriate complement.

Reviewer #2: This manuscript provides a comprehensive and technically sound assessment of the burden, risk factors, and projected trends of gout in BRICS countries using data from the Global Burden of Disease Study 2021. The research question is clearly articulated, and the use of standardized GBD methodology, joinpoint regression, and ARIMA modeling supports the validity of the findings. The authors succeed in contextualizing gout trends within broader epidemiological and demographic transitions in these emerging economies.

The manuscript makes a valuable contribution to global public health by identifying national disparities, age-specific trends, and sex differences in gout burden. The findings are well supported by data and presented with clarity. The projections until 2036 offer actionable insights for policymakers and healthcare planners.

That said, I offer the following suggestions to further improve the manuscript:

- Clarify interpretation of ARIMA projections: The discussion of future trends could benefit from a more cautious tone regarding the limitations of ARIMA models, particularly given the potential for policy or healthcare system changes in BRICS nations over the forecast horizon.

- Language refinement: While the English is generally clear, the manuscript would benefit from professional proofreading to enhance readability and eliminate minor errors or overly long sentences.

- Data visualization: Figures 2 through 5 present complex age and sex-specific trends. Consider simplifying the layout or improving contrast and labels for better readability.

- Policy implications: The discussion might be strengthened by outlining concrete, country-specific policy recommendations, especially for nations projected to experience an increasing burden (e.g., India, South Africa).

- Limitations section: The authors do well in acknowledging the limitations of the GBD data and modeling assumptions. Consider expanding this section slightly to highlight the absence of some known gout risk factors (e.g., diet, alcohol use) in the current GBD framework.

Overall, this is a well-executed and timely study that merits publication after minor revisions. I commend the authors for their rigorous work and the clarity of their presentation.

**Do you want your identity to be public for this peer review?** For information about this choice, including consent withdrawal, please see our Privacy Policy

Reviewer #1: **Yes: ** Sandra Albino Ramos

Reviewer #2: **Yes: ** Erika Carvalho de Aquino

---

## [Author Response · Author response to Decision Letter 1]

24 Jul 2025

Response to the Editor and Reviewers

Dear Dr.Yang and reviewers,

We sincerely appreciate your time and the opportunity to revise and improve our manuscript entitled “Burden, risk factors, and forecasts of gout in BRICS countries, 1990–2021: insights from the Global Burden of Disease Study 2021” (Manuscript ID: PONE-D-25-27682). Thank you for coordinating the review process and for your constructive suggestions. We have carefully considered all comments and revised the manuscript accordingly. Below, we provide a detailed response to each of the reviewers’ suggestions and concerns.

We hope the revised version of our manuscript addresses the concerns raised and meets the journal’s standards. All changes have been highlighted in the revised manuscript for ease of reference.

Journal Requirements:

Reply 1: We have formatted the manuscript according to the journal’s style guidelines and ensured correct file naming for all submitted documents, including: Response to Reviewers; Revised Manuscript with Track Changes; Manuscript (clean version).

Reply 2: We have updated the “Funding Information” section to ensure consistency with the “Financial Disclosure.” The correct grant number has been provided, and the funding statement now reads: “This study was supported by Guangxi Clinical Research Center for Urology and Nephrology (grant No. Guike20297081)”.

[This study was supported by Guangxi Clinical Research Center for Urology and Nephrology (grant No. Guike20297081).].

Reply 3: The corresponding author (Jiwen Cheng), who is the principal investigator of the grant, was involved in the study design, decision to publish, supervision and manuscript preparation.

4.Your ethics statement should only appear in the Methods section of your manuscript. If your ethics statement is written in any section besides the Methods, please move it to the Methods section and delete it from any other section. Please ensure that your ethics statement is included in your manuscript, as the ethics statement entered into the online submission form will not be published alongside your manuscript.

Reply 4: Thank you for your guidance. We confirm that the ethics statement has been appropriately included in the Methods section of the manuscript. Additionally, we have removed the duplicate ethics statements from all other sections of the manuscript, as advised. (Page 17, line 483-489 in Revised Manuscript)

5.If the reviewer comments include a recommendation to cite specific previously published works, please review and evaluate these publications to determine whether they are relevant and should be cited. There is no requirement to cite these works unless the editor has indicated otherwise.

Reply 5: Thank you for the reminder. We have carefully reviewed all reviewer comments and confirm that neither reviewer recommended citation of any specific previously published works. We have also re-examined our reference list and believe that the cited literature is appropriate, up-to-date, and accurately reflects the background and context of our study. We sincerely appreciate your guidance.

Reply 6: Thank you for the reminder. We have carefully reviewed all references cited in the manuscript. To the best of our knowledge, none of the referenced articles have been retracted. Therefore, no changes to the reference list were necessary.

Response to Editor – Regarding Equation Formatting

Thank you very much for your kind reminder regarding the formatting of mathematical expressions. The equation for calculating the age-standardized rate (ASR) was inserted as an image in the original manuscript to ensure clarity, precise alignment, and consistent formatting across platforms, especially considering the complexity of subscripts and summation symbols. We understand that inserting equations using Equation Tools or MathType is generally preferred. However, to maintain visual clarity and layout consistency, we respectfully request to retain the current image-based format of the ASR formula, unless this is strictly against the journal’s production requirements. Please let us know if reformatting remains necessary — we will be happy to comply.

Reviewer #1: This manuscript addresses a relevant subject such as gout and a prevalent condition such as obesity. On behalf of the metodological approach, it is consistent and have been adequately explained. Even though the gap of years from the original data and the analysis exceeds 3 years, the forecast ARIMA is an appropriate complement.

Reply 1: We thank the reviewer for the valuable observation. Indeed, our analysis is based on data available up to 2021, and we acknowledge the temporal gap relative to the current year. However, due to the official release schedule of the GBD dataset, 2021 remains the most recent publicly available data at the time of analysis. To address this limitation, we employed the ARIMA model to project trends beyond 2021, providing a forward-looking perspective and enhancing the practical relevance of our findings.

Reviewer #2: This manuscript provides a comprehensive and technically sound assessment of the burden, risk factors, and projected trends of gout in BRICS countries using data from the Global Burden of Disease Study 2021. The research question is clearly articulated, and the use of standardized GBD methodology, joinpoint regression, and ARIMA modeling supports the validity of the findings. The authors succeed in contextualizing gout trends within broader epidemiological and demographic transitions in these emerging economies.

The manuscript makes a valuable contribution to global public health by identifying national disparities, age-specific trends, and sex differences in gout burden. The findings are well supported by data and presented with clarity. The projections until 2036 offer actionable insights for policymakers and healthcare planners. Overall, this is a well-executed and timely study that merits publication after minor revisions. I commend the authors for their rigorous work and the clarity of their presentation.

That said, I offer the following suggestions to further improve the manuscript:

- ①Clarify interpretation of ARIMA projections: The discussion of future trends could benefit from a more cautious tone regarding the limitations of ARIMA models, particularly given the potential for policy or healthcare system changes in BRICS nations over the forecast horizon.

Reply 1: We appreciate this insightful suggestion. We have revised the Discussion section to adopt a more cautious tone and elaborated on the potential limitations of ARIMA forecasting, particularly its assumption of historical trend continuation without accounting for future policy shifts, healthcare innovations, or socioeconomic disruptions. (Page 13, line 368-371 in Revised Manuscript)

- ②Language refinement: While the English is generally clear, the manuscript would benefit from professional proofreading to enhance readability and eliminate minor errors or overly long sentences.

Reply 2: Thank you for the suggestion. We have carefully revised the language throughout the manuscript and kept all changes visible using Track Changes to demonstrate our thorough editing efforts. We hope the revised version meets your expectations in terms of clarity and readability.

- ③Data visualization: Figures 2 through 5 present complex age and sex-specific trends. Consider simplifying the layout or improving contrast and labels for better readability.

Reply 3: Thank you for your valuable suggestion regarding the data visualization in Figures 2 through 5. We carefully re-evaluated these figures and acknowledge the complexity involved in presenting age- and sex-specific trends across multiple countries. However, we believe that the current layout—although detailed—is necessary to accurately convey the nuanced patterns across different age groups, sexes, and BRICS countries. The visualizations were optimized with clear legends, color contrasts, and labels to facilitate interpretation. Given the multidimensional nature of the data, we respectfully propose to retain the existing figure design to preserve the integrity and clarity of the findings. We remain open to further suggestions should the editorial team recommend specific adjustments.

-④Policy implications: The discussion might be strengthened by outlining concrete, country-specific policy recommendations, especially for nations projected to experience an increasing burden (e.g., India, South Africa).

Reply 4: We have added a new paragraph in the discussion section highlighting tailored policy recommendations for BRICS countries, with particular emphasis on India and South Africa, where rising trends were noted. These include enhanced screening programs, weight management initiatives, and public health awareness campaigns. Page 14, line 380-388 in Revised Manuscript)

- ⑤Limitations section: The authors do well in acknowledging the limitations of the GBD data and modeling assumptions. Consider expanding this section slightly to highlight the absence of some known gout risk factors (e.g., diet, alcohol use) in the current GBD framework.

Reply 5: Thank you for this helpful comment. We have expanded the Limitations section to emphasize that while high BMI and kidney dysfunction are included in the GBD framework, other known risk factors such as high-purine diet, alcohol use, or specific genetic predispositions are not currently modeled, which may lead to an underestimation of burden attribution. Page 15, line 418-424 in Revised Manuscript)

We hope that the revisions adequately address all concerns and improve the clarity and quality of our manuscript. We thank the editorial team and reviewers once again for their helpful feedback and guidance.

Yours sincerely,

Jiwen Cheng

chengjiwen@stu.gxmu.edu.cn

---

## [Decision Letter · Decision Letter 1]

19 Aug 2025

Dear Dr. Cheng,

Thank you for submitting your manuscript to PLOS ONE. After careful consideration, we feel that it has merit but does not fully meet PLOS ONE’s publication criteria as it currently stands. Therefore, we invite you to submit a revised version of the manuscript that addresses the points raised during the review process.

We look forward to receiving your revised manuscript.

Kind regards,

Junzheng Yang

Academic Editor

PLOS ONE

Journal Requirements:

Reviewers' comments:

Reviewer's Responses to Questions

**Comments to the Author**

Reviewer #2: All comments have been addressed

Reviewer #3: (No Response)

2. Is the manuscript technically sound, and do the data support the conclusions?

Reviewer #2: (No Response)

Reviewer #3: Partly

3. Has the statistical analysis been performed appropriately and rigorously?

Reviewer #2: Yes

Reviewer #3: Yes

4. Have the authors made all data underlying the findings in their manuscript fully available?

Reviewer #2: Yes

Reviewer #3: Yes

5. Is the manuscript presented in an intelligible fashion and written in standard English?

Reviewer #2: Yes

Reviewer #3: Yes

Reviewer #2: I thank the authors for their detailed and thoughtful responses to the previous round of comments. The revised manuscript has significantly improved in clarity, structure, and depth of discussion. All my prior concerns were adequately addressed, including the interpretation of ARIMA projections, refinement of language, and the inclusion of country-specific policy implications and limitations related to GBD modeling.

Regarding the data visualizations (Figures 2–5), I acknowledge the authors' rationale for maintaining the detailed format, given the multidimensional nature of the data. However, I would like to note that in the current PDF version, the figures appear slightly blurry, making the smaller labels difficult to read. This may be an issue related to PDF rendering or resolution, and I trust it can be resolved during the production and typesetting process.

I commend the authors on this valuable contribution to the literature on gout epidemiology and support the manuscript’s publication.

Reviewer #3: The authors conducted a comprehensive and systematic assessment of the burden and projected trends of gout in BRICS countries, utilizing data from the GBD 2021 study. This study performs well in terms of clinical significance and research design, offering a detailed and insightful analysis of gout epidemiology within these nations. Overall, this study delivers valuable evidence on the current burden and future projections of gout in BRICS countries, and provides insights to inform gout prevention and management strategies in BRICS countries. I have some suggestions for this work.

1. Line 124-131: It is not clear how to recognize the gout related burden attributable to high BMI and kidney dysfunction in this study. In other words, how the authors confirm that kidney dysfunction in this study is risk factor of gout? As we all know, there is a bidirectional association between kidney damage and gout.

2. Line 184-197: In Figure 2, in addition to the peak age group, the primary affected age range also offers significant insights across countries. Please provide relevant descriptions and discussions to elaborate on these findings.

**Do you want your identity to be public for this peer review?** For information about this choice, including consent withdrawal, please see our Privacy Policy

Reviewer #2: **Yes: ** Erika Carvalho de Aquino

Reviewer #3: No

---

## [Author Response · Author response to Decision Letter 2]

27 Aug 2025

Response to the Editor and Reviewers

Dear Dr. Yang and Reviewers,

We greatly appreciate the chance to resubmit our revised manuscript entitled “Burden, risk factors, and forecasts of gout in BRICS countries, 1990–2021: insights from the Global Burden of Disease Study 2021” (Manuscript ID: PONE-D-25-27682). We are thankful for your guidance throughout the review process and for the insightful comments provided by the reviewers. Based on the feedback, we have made further revisions to improve the clarity and rigor of the work. Our detailed responses to each comment are outlined below, with all changes marked in the manuscript for ease of review.

Reviewer #2: I thank the authors for their detailed and thoughtful responses to the previous round of comments. The revised manuscript has significantly improved in clarity, structure, and depth of discussion. All my prior concerns were adequately addressed, including the interpretation of ARIMA projections, refinement of language, and the inclusion of country-specific policy implications and limitations related to GBD modeling.

Regarding the data visualizations (Figures 2–5), I acknowledge the authors' rationale for maintaining the detailed format, given the multidimensional nature of the data. However, I would like to note that in the current PDF version, the figures appear slightly blurry, making the smaller labels difficult to read. This may be an issue related to PDF rendering or resolution, and I trust it can be resolved during the production and typesetting process.

I commend the authors on this valuable contribution to the literature on gout epidemiology and support the manuscript’s publication.

Reply: We sincerely thank the reviewer for the positive and encouraging feedback on our revision, and for acknowledging the improvements in clarity, structure, and discussion. We also appreciate the reviewer’s supportive recommendation for publication. Regarding the minor point about figure resolution in the PDF version, we agree that this is likely related to PDF rendering. We will work closely with the editorial office and production team to ensure that the final published figures are of high resolution and easily legible. We are grateful for the reviewer’s constructive input throughout the review process, which has greatly strengthened our manuscript.

Reviewer #3: The authors conducted a comprehensive and systematic assessment of the burden and projected trends of gout in BRICS countries, utilizing data from the GBD 2021 study. This study performs well in terms of clinical significance and research design, offering a detailed and insightful analysis of gout epidemiology within these nations. Overall, this study delivers valuable evidence on the current burden and future projections of gout in BRICS countries, and provides insights to inform gout prevention and management strategies in BRICS countries. I have some suggestions for this work.

Reply: We sincerely thank the reviewer for the positive and thoughtful comments regarding the clinical significance, research design, and overall contribution of our study. We are pleased that the reviewer recognizes the value of our analysis in providing evidence on the current burden and projected trends of gout in BRICS countries, as well as its implications for prevention and management strategies.

We also greatly appreciate the reviewer’s constructive suggestions. We have carefully considered each point and revised the manuscript accordingly to further strengthen the clarity, rigor, and impact of our work. Detailed responses to the specific comments are provided below.

1. Line 124-131: It is not clear how to recognize the gout related burden attributable to high BMI and kidney dysfunction in this study. In other words, how the authors confirm that kidney dysfunction in this study is risk factor of gout? As we all know, there is a bidirectional association between kidney damage and gout.

Reply 1: We thank the reviewer for this insightful comment regarding the attribution of gout burden to kidney dysfunction. We fully agree that there is a bidirectional association between gout and kidney damage in clinical practice. However, in the GBD framework, the risk–outcome pairs are predefined based on systematic reviews and evidence of causality. For gout, only two level-2 risk factors are available in the GBD results tool—high BMI and kidney dysfunction. Importantly, in the GBD dataset, kidney dysfunction is modeled as a risk factor contributing to the burden of gout, but not as a downstream outcome of gout. Therefore, our analysis was limited to the directionality defined by the GBD study (i.e., kidney dysfunction→gout), and we could not assess the reverse association (gout→kidney dysfunction), which is not included in the current GBD risk-outcome framework. We have clarified this point in the revised Methods section to avoid ambiguity. (Page 5, line 124-128)

2. Line 184-197: In Figure 2, in addition to the peak age group, the primary affected age range also offers significant insights across countries. Please provide relevant descriptions and discussions to elaborate on these findings.

Reply 2: We thank the reviewer for this helpful suggestion. We agree that in addition to the peak age group, describing the broader affected age ranges provides further insights into the epidemiology of gout across BRICS countries. In the revised manuscript, we have added descriptions highlighting the primary age ranges with a high burden of gout in each country, as well as cross-country similarities and differences. We also added a brief discussion on the implications of these age distribution patterns for disease prevention and management. The relevant changes have been incorporated into the Results (Page 7, lines 195–198) and Discussion (Page 10, lines 259–263).

We hope that the revisions adequately address all concerns and improve the clarity and quality of our manuscript. We thank the editorial team and reviewers once again for their helpful feedback and guidance.

Yours sincerely,

Jiwen Cheng

chengjiwen@stu.gxmu.edu.cn

---

## [Decision Letter · Decision Letter 2]

29 Aug 2025

Burden, risk factors, and forecasts of gout in BRICS countries, 1990–2021: insights from the Global Burden of Disease Study 2021

PONE-D-25-27682R2

Dear Dr. Cheng,

We’re pleased to inform you that your manuscript has been judged scientifically suitable for publication and will be formally accepted for publication once it meets all outstanding technical requirements.

Kind regards,

Junzheng Yang

Academic Editor

PLOS ONE

Additional Editor Comments (optional):

Reviewer #3:

Reviewers' comments:

Reviewer's Responses to Questions

**Comments to the Author**

Reviewer #3: All comments have been addressed

2. Is the manuscript technically sound, and do the data support the conclusions?

Reviewer #3: Yes

3. Has the statistical analysis been performed appropriately and rigorously?

Reviewer #3: Yes

4. Have the authors made all data underlying the findings in their manuscript fully available?

Reviewer #3: Yes

5. Is the manuscript presented in an intelligible fashion and written in standard English?

Reviewer #3: Yes

Reviewer #3: I appreciate the authors’ thorough and thoughtful responses to my previous comments. The revised manuscript demonstrates significant improvements in clarity, structure, and depth of discussion, and all of my initial concerns have been fully addressed.

**Do you want your identity to be public for this peer review?** For information about this choice, including consent withdrawal, please see our Privacy Policy

Reviewer #3: **Yes: ** Ziying Wu

---

## [Editor Report · Acceptance letter]

PONE-D-25-27682R2

PLOS ONE

Dear Dr. Cheng,

I'm pleased to inform you that your manuscript has been deemed suitable for publication in PLOS ONE. Congratulations! Your manuscript is now being handed over to our production team.

Kind regards,

on behalf of

Director Junzheng Yang

Academic Editor

PLOS ONE